# The Role of Chemotactic Cytokines in Tick-Borne Encephalitis

**DOI:** 10.3390/cells14070490

**Published:** 2025-03-25

**Authors:** Sambor Grygorczuk, Piotr Czupryna, Diana Martonik, Anna Parfieniuk-Kowerda, Justyna Adamczuk, Justyna Dunaj-Małyszko, Maciej Giecko, Joanna Osada, Miłosz Parczewski, Robert Flisiak, Anna Moniuszko-Malinowska

**Affiliations:** 1Department of the Infectious Diseases and Neuroinfections, Medical University in Białystok, 15-089 Białystok, Poland; piotr.czupryna@umb.edu.pl (P.C.); justyna.dunaj@umb.edu.pl (J.D.-M.); maciej.giecko@umb.edu.pl (M.G.); anna.moniuszko@umb.edu.pl (A.M.-M.); 2Department of the Infectious Diseases and Hepatology, Medical University in Białystok, 15-089 Białystok, Poland; diana.martonik@umb.edu.pl (D.M.); anna.parfieniuk@umb.edu.pl (A.P.-K.); robert.flisiak@umb.edu.pl (R.F.); 3University Hospital in Białystok, 15-276 Białystok, Poland; justyna.adamczuk@umb.edu.pl; 4Department of the Hematologic Diagnostics, Medical University in Białystok, 15-089 Białystok, Poland; joanna.osada@umb.edu.pl; 5Department of the Infectious Diseases, Tropical Diseases and Acquired Immunodeficiencies, Pomeranian Medical University, 70-204 Szczecin, Poland; mparczewski@yahoo.co.uk

**Keywords:** tick-borne encephalitis, cerebrospinal fluid, intrathecal immunity, pleocytosis, chemotaxis, chemokines

## Abstract

In tick-borne encephalitis (TBE), the central nervous system (CNS) is infiltrated by a mixed leukocyte population contributing both to the infection control and the immune-mediated pathology. To elucidate the roles of chemotactic cytokines in this process, we measured concentrations of 25 cytokines in serum and cerebrospinal fluid (CSF) simultaneously with total CSF leukocyte count (pleocytosis) and leukocyte subpopulation counts in 103 TBE patients. We created models describing the dependence of pleocytosis and clinical severity on cytokine concentrations. Ten polymorphisms in genes for cytokines or their receptors were studied with rtPCR in patients’ DNA samples. The strongest chemotactic gradients towards CSF were created by CXCL1, IL-8, CXCL10, CCL2, CCL3, CCL4, CCL7, CCL8, CCL19 and CCL20. Neutrophil counts in CSF correlated with concentrations of CXCL1 and IL-8 and lymphocyte counts with IL-16, CCL19, CCL20, CCL4, CXCL12, and CXCL13. The milder disease is associated with CCL11, CCL19, CXCL10 and CXCL13,-while the more severe with CXCL1 and CCL20. The polymorphisms in the genes *CCR2*, *CCL5*, *CXCR3* and *CX3CR1* are associated with the cytokine concentrations and pleocytosis, but not with clinical severity. Multiple chemotactic cytokines contribute to pleocytosis in TBE, with no straightforward relationship between their effects on pleocytosis and the clinical presentation.

## 1. Introduction

The central nervous system (CNS) is an immunologically privileged site; in normal conditions, it is patrolled only by a small population of lymphoid cells, mostly T lymphocytes [1]. In CNS infections, the influx of the immune cells, indispensable for the infection control, occurs across the blood–brain barrier (BBB) in a controlled manner and is clinically detectable as the cerebrospinal fluid (CSF) pleocytosis [1,2,3,4]. The extent and composition of the CSF leukocyte population differ, depending on the etiology and severity of the infection and provide an important diagnostic criterion in CNS infectious diseases [5,6,7,8]. In *Flavivirus* tick-borne encephalitis (TBE), a variable and temporally evolving population of leukocytes is detectable in CSF, typically dominated by T CD4+ lymphocytes with a significant contribution of T CD8+ lymphocytes and neutrophils [9,10,11]. The animal data from *Flavivirus* encephalitis models [3,4,12,13] and from the human autopsy studies [14] suggest that different leukocyte subsets may play particular protective or immunopathogenic roles in the intrathecal immune response. In human TBE, higher pleocytosis tends to associate with a more severe neurologic involvement and the counts of different leukocyte populations with particular disease presentation, supporting the notion of their variable pathogenetic roles [11]. However, the roles of and interactions between the leukocyte subsets and the mechanisms of their migration into CNS are not well understood. Elucidating this process could potentially improve the treatment of TBE, which is currently purely symptomatic and supportive, while no interventions specifically directed at the pathogenesis of the disease are available [15].

Chemokines are a family of over 30 chemotactic cytokines acting through specific membrane receptors, indispensable both for the maturation and circulation of the immune cells in health and for their migration towards the inflammatory focus [16]. Because of the chemokine ligand-receptor interaction specificity and the variable expression of chemokine receptors by different leukocyte populations, chemokines allow for a precise regulation of a composition of an inflammatory infiltrate [16]. Chemokines may be divided into several groups sharing the receptor and cellular affinity. The CXC-chemokine (alpha-chemokine) subfamily includes, among others, the agonists of CXCR1 and/or CXCR2 receptors, from CXCL1 to CXCL8 (IL-8). They are chemotactic proteins for neutrophils but may also attract mononuclear leukocytes expressing CXCR1 [16]. Another group of CXC-chemokines, acting through the CXCR3 receptor (CXCL9, CXCL10 and CXCL11), attracts lymphocytes and is associated with Th1-type immunity [16]. CXCL12 and CXCL13 are distinct CXC-chemokines acting through their specific receptors CXCR4 and CXCR5, respectively [16]. CXCL12 is a wide-spectrum chemokine for mononuclear cells, while CXCL13 is a more specific attractant of B and T CD8+ lymphocytes [16]. Within the CC-chemokine (beta-chemokine) subfamily, CCL2 is a main ligand for CCR2 expressed by monocytes, activated Th2 lymphocytes and NK cells, and is considered crucial in regulating the monocyte migration [16,17,18]. The related chemokines, CCL7, CCL8, CCL11 and CCL13, act with different specificities through CCR1, CCR2 and/or CCR3 on both monocytes and various lymphoid cell populations [16]. CCL3, CCL4 and CCL5 share a common CCR5 receptor expressed by Th1, Th17, NK cells and monocytes [16,19]. CCL17 is the specific agonist of CCR4 expressed by Th2, Th17 and Treg lymphocytes. CCL20 is the agonist of CCR6 expressed mainly by Th17 and Treg cells [16,20,21]. The simultaneous expression of CCR4 and CCR6 is especially characteristic for Th17 lymphocytes [20]. The CCR7 agonists CCL19 and CCL21 are involved in the T lymphocyte maturation and homeostasis but have recently been recognized as mediators of the lymphocyte chemotaxis in inflammation as well [16]. Finally, CX3CL1 (fractalkine) is a unique chemokine expressed primarily in a transmembrane form, including on the luminal surface of endothelia, from which it may be proteolytically cleaved; its receptor CX3CR1 is expressed by mononuclear leukocytes [16]. Additional cytokines not belonging to the chemokine family may exert chemotactic effects, especially IL-16, a pro-inflammatory cytokine signaling through CD4 receptor and chemotactic for T CD4+ lymphocytes and for T CD8+ lymphocytes which have upregulated CD4 upon stimulation [22].

It has been suggested that the lymphocyte migration into CNS in *Flavivirus* encephalitis is mediated mainly by CCR5-agonist chemokines (CCL3, CCL4 and/or CCL5) with contribution from CXCR3 agonists (CXCL9, CXCL10 and/or CXCL11), as observed in animal models [13,17,19,23,24,25,26] and supported by some human genotyping studies [27,28,29,30] and clinical data [10,31]. However, the later results suggest that in human TBE, CCR5 expression is not indispensable for TBEV control [32], while other chemokines, like CCL19, are highly upregulated in CNS of TBE patients and correlate with the CSF cytometric parameters [33]. Further complicating the picture, IL-16, a pro-inflammatory and chemotactic cytokine not belonging to the chemokine family, has been detected in the CSF of patients with viral neuroinfections, including TBE, in high concentrations correlating with CSF pleocytosis [31,34].

We have studied the expression of a panel of 25 chemotactic cytokines in a large group of TBE patients to define which chemotactic factors are upregulated intrathecally and may contribute to CSF pleocytosis, with a focus on lymphocyte counts.

## 2. Materials and Methods

### 2.1. Study Group

In the course of the research project “Factors influencing presentation and outcome of tick-borne encephalitis”, adult TBE patients hospitalized in the Department of the Infectious Diseases and Neuroinfections of the Medical University in Białystok between 2019 and 2023 were recruited and studied simultaneously for multiple biochemical, cellular and genetic markers potentially related to the pathogenesis and clinical course of TBE. The patients included in the TBE group (1) had a history of a tick bite or an exposition to ticks in an endemic area within 3 weeks before the onset of symptoms, (2) presented with an acute febrile disease with symptoms suggestive of meningitis and/or encephalitis, (3) had a CSF pleocytosis ≥ 5/μL in a diagnostic lumbar puncture on admission as a hallmark of the CNS inflammation [8] and (4) had specific anti-TBEV IgM antibodies detected in CSF on admission or seroconverted in serum and/or CSF on follow-up, fulfilling the European criterion for the confirmed TBE case [35]. The patients fulfilling criteria (2) and (3) consistently seronegative towards TBEV both on admission and on follow-up were considered having a viral meningitis and/or encephalitis of non-TBEV etiology and included in an aseptic meningitis (AM) reference group. No TBE or AM patients had any probable alternative diagnosis of the meningo (encephalitis), coexisting active CNS disease or severe coinfection (e.g., HIV or active purulent bacterial infection). In both TBE and AM groups, patients with no focal neurologic deficits or altered consciousness were classified as having uncomplicated meningitis (M) and patients with an altered mental status and/or focal neurologic symptoms as having meningoencephalitis (ME) or meningoencephalomyelitis (MEM). In the TBE group, the disease severity was additionally assessed quantitatively with an objective scoring scale described and validated previously [36].

The control groups included (1) for CSF (non-inflammatory CSF samples from hospitalized patients in whom CNS infection had been excluded by means of a lumbar puncture) and (2) for serum (blood samples from healthy volunteers (blood donors)).

All the patients and healthy volunteers gave informed written consent for participation in the study.

### 2.2. Material Acquisition and Handling

The paired blood and CSF samples were obtained on admission to hospital, together with the material for clinically indicated diagnostic examinations and stored in 5 °C for no more than 24 h before processing. For the cytokine detection, 1.0 mL CSF sample was collected into a sterile plastic tube and 2.6 mL blood sample was collected on clot, centrifuged within an hour and serum pipetted into a fresh vial. The serum and CSF samples were frozen on the day of collection, stored at −40 °C and thawed directly before use at the end of the study period. For flow cytometry, a 1.0 mL sample of CSF was collected into a sterile plastic tube and a 1.4 mL sample of blood into a 1.4 mL EDTA-coated tube and analyzed cytometrically within 24 h.

For DNA isolation and genotyping, an additional blood sample was taken into a 1.4 mL EDTA-coated tube.

### 2.3. Basic Laboratory Examinations

Anti-TBEV IgM and IgG antibodies were detected with EuroImmun Anti-TBE virus ELISA kits (Lübeck, Germany) in serum and CSF samples obtained on admission, as well as on follow-up in originally seronegative patients, following the standard procedure in a hospital diagnostic laboratory. The sensitivity and specificity of the serum anti-TBEV IgM assay is reported by the manufacturer to be 100% (based on evaluation in a population of 129 patients). The negative results on admission were verified on follow-up before discharge or during control visits and only if they remained negative patients were definitively assigned to a non-TBE meningitis group.

The CSF pleocytosis with differential and albumin concentrations in serum and CSF were measured with standard laboratory techniques as part of the routine diagnostics.

### 2.4. Cytokine Concentrations

The concentrations of the soluble factors were measured simultaneously with a micro-bead technique with Luminex assays (R&D Systems, Minneapolis, MN, USA) in paired serum and cerebrospinal fluid samples. The whole set of soluble mediators included 84 cytokines, soluble receptors, metalloproteinases, soluble forms of adhesion molecules and brain-derived molecules, measured with a set of 7 customized microbead assays, constructed for the purpose of the study from the analytical panel offered by Luminex. These included 24 cytokines selected for this analysis on the basis of their chemotactic activity towards different populations of leukocytes: IL-16, alpha-chemokines (CXC-subfamily chemokines) CXCL1, CXCL2, CXCL5, CXCL6, IL-8, CXCL9, CXCL10, CXCL11 and CXCL13, beta-chemokines (CC-subfamily chemokines): CCL2, CCL3, CCL4, CCL5, CCL7, CCL8, CCL11, CCL13, CCL17, CCL19, CCL20 and CCL21, fractalkine (CX3CL1) and lymphokine (XCL1). The measurements were performed according to the manufacturer’s instruction on a Bio-Plex 200 System (Bio-Rad Laboratories, Hercules, CA, USA). The measurement of the concentration of alpha-chemokine CXCL12 was not available with the Luminex microbead system at the time of the study and as a consequence it was measured with a commercial ELISA kit from R&D Systems (Minneapolis, MN, USA), following the manufacturer’s instructions. These measurements were carried out in CSF only, as the assay was not certified for use on serum samples. The concentration of each cytokine was expressed in pg/mL. The values below the detection limit were considered 0.

### 2.5. Flow Cytometry

The blood samples were treated with Lysing Solution to lyse erythrocytes. Both blood and CSF samples were washed with PBS directly before the flow cytometry. BD Multitest 6-color TBNK Reagent kit (BD Biosciences, San Jose, CA, USA) was used to study peripheral blood and CSF lymphocyte fractions, strictly following the manufacturer’s instructions. The test applied a cocktail of multiple fluorescently labeled monoclonal antibodies: CD45-PerCP-Cy5.5, CD3-FITC, CD19-APC, CD16/CD56-PE, CD4-PE-Cy7, and CD8-APC-Cy7, allowing to determine fractions of mature T, B and NK lymphocytes as well as helper/inducer and suppressor/cytotoxic T cell subset ratios in a single tube. The procedure was performed on FACS Canto II (BD Biosciences) cytometer, and the read-outs were analyzed using BD FACS Canto clinical software. The example of a read-out in a representative pair of blood and CSF samples is presented in Figure A1 in Appendix B.

### 2.6. Genotyping

Genomic DNA was extracted from peripheral blood with QIAamp DNA Blood Mini Kit (QIAgen, Hilden, Germany), re-suspended in 200 µL of AE buffer (QIAgen) and stored at 4 °C.

TaqMan SNP (Applied Biosystems/Life Technologies, Foster City, CA, USA) genotyping assays were used following the manufacturer’s protocol with real-time PCR technique on the StepOne thermal cycler (Applied Biosystems/Life Technologies). The genotypes were identified with TaqMan Genotyper Software v1.0.1 (Applied Biosystems/Life Technologies). The common single nucleotide polymorphisms (SNPs) in genes for chemokines and chemokine receptors belonging to four pathogenetically important chemotactic signaling axes were analyzed as follows: (1) CCR2-related: rs1799864 in *CCR2*; (2) CCR5-related: rs2107538, rs2280788 in *CCL5*; rs1799987 in CCR5; (3) CXCR3-related: rs2280964 and rs34334103 in *CXCR3* (located on X chromosome), rs4508917 in *CXCL10*; rs6817952 in *CXCL11*; (4) CX3CR1-related: rs3732379 in *CX3CR1*. Additionally, *CCR5Δ32* deletion was analyzed with PCR with sequence specific primers, as described previously [37]. Visualization under UV light was performed after electrophoresis on the 2.5% agarose gel (SIGMA, Saint Louis, MO, USA) stained with DNA-star dye (Lonza Inc., Rockland, MI, USA).

### 2.7. Data Analysis

We have calculated CSF/serum concentration gradients and CSF concentration indexes I_cytokine_ for all the cytokines except of CCL5, for which the serum concentrations tend to be overestimated because of the release from thrombocytes during the sample handling [38], and CXCL12, for which the serum concentrations were not evaluated.

The likelihood of the intrathecal synthesis of the cytokines, as opposed to the leakage from plasma to CSF through the impaired BBB, was evaluated by calculating the CSF concentration index according to the following formula:I_cytokine_ = (CSF_cytokine_/serum_cytokine_)/CSF_albumin_/serum_albumin_.(1)

The cut-off value defining the intrathecal synthesis of a macromolecular compound depends on its molecular weight (MW), approximating 1 if the MW of the compound is equal to MW of serum albumin [39] and is not standardized specifically for chemotactic cytokines. Even corrected for MW, the formula is a linear approximation of the non-linear relation between the BBB permeability to different classes of proteins, becoming less reliable with increasing BBB permeability, and the result should be considered an estimate only [8]. We assumed values >> 1 as consistent with the intrathecal synthesis and values on the order of 1 or lower as effectively excluding it.

The concentration gradient was defined as CSFc_ytokine_/serum_cytokine_, with values > 1 pointing to a concentration gradient towards CSF, consistent with a cytokine attracting its target cells from blood into CSF compartment, and <1 to a gradient towards periphery. The CSF concentration gradient > 1 can also be considered as proof of the intrathecal synthesis [8].

If a cytokine was undetectable in CSF, the concentration gradient and index were assumed 0. For the undetectable serum concentration, the CSF/serum gradient was calculated with the conservatively assumed serum concentration of a half of the lowest detected value.

The flow cytometry results were originally read out as fractions within the lymphoid population and recalculated to the absolute cell numbers from these fractions and the absolute CSF lymphocyte count.

For the analysis, the ME and MEM patients were pooled into a single ME/MEM subgroup of TBE patients with neurologic involvement, as opposed to an uncomplicated meningitis (M) subgroup. The data were analyzed with STATISTICA 13 (TIBCO Software Inc., Palo Alto, CA, USA) software with non-parametric tests: non-parametric Kruskal–Wallis ANOVA for multiple group comparisons, Mann–Whitney test for two-group comparisons, Wilcoxon pair test for two-variable comparisons and chi-square test for correlation analysis were used. *p* < 0.05 was considered statistically significant. To verify and extend the findings from the non-parametric analyses, multiple regression models were constructed with the CSF cytometric parameters as dependent variables and chemokine CSF concentrations and concentration gradients as independent variables. The models were built for the total pleocytosis, the lymphocyte count and the main lymphocyte populations following the same procedure. We included both chemokine concentrations and concentration gradients attempting to distinguish the purely chemotactic effect (best described by a concentration gradient) from the alternative mechanisms linking chemokine concentration with a cell count (e.g., intrathecal synthesis of a chemokine by infiltration leukocytes) better reflected by an intrathecal concentration expressed in absolute terms. We did not include chemokines that were not upregulated in CSF (CCL5, XCL1) and did not include the gradient and concentration of the same chemokine in one model, as they were linearly inter-correlated by definition. Chemokines with evidently non-linear correlation on visual inspection were either excluded from the analysis (CCL17) or included with the exclusion of the individual observations causing non-linearity (e.g., CXCL1, IL-8). The strongly inter-correlated variables with a tolerance coefficient < 0.1 were excluded and the remaining variables were used as an input to the stepwise regression analysis, with the resulting models tested and refined manually. The models with an unrealistic value of a free parameter (tending towards negative or significantly above 0) were rejected or further refined.

The stepwise multiple linear regression model was built with an analogous procedure to verify associations of chemokine concentrations and gradients as independent variables with the disease severity score as a dependent variable.

Genetic polymorphisms were analyzed with non-parametric tests for the associations between the genotypes and (1) the expression of the relevant chemokines: CCL2, CCL7, CCL8 and CCL13 for rs1799864 in *CCR2*; CCL3, CCR4 and CCL5 for polymorphisms in CCR5 axis-related genes; CXCL9, CXCL10 and CXCL11 for SNPs in CXCR3 axis-related genes; CX3CL1 for rs3732379 in *CX3CR1*; as well as (2) total pleocytosis and CSF counts of the mononuclear cell populations and (3) clinical severity score.

## 3. Results

### 3.1. Study Group

The study group consisted of 103 adult patients with a serologically confirmed TBE (66 male, 37 female, median age 47 years), including 53 with M (36 male, 17 female, median age 44 years), 45 with ME (26 male, 19 female, median age 51 years) and 5 with MEM (4 male, 1 female, median age 52 years). The tendency for a higher age in more severe presentations was not statistically significant. Due to a limited CSF sample volume and sample processing limitations, individual chemokine concentration measurements were not available in 5 patients (3 male, 2 female, all with ME), the genotyping was not performed in 2 patients (both female with ME) and a full CSF cytometric analysis, including lymphocyte subsets, was performed in 56 patients (median age 44.5 year, 36 male and 20 female, 26 with M, 27 with ME, 3 with MEM, genetic analysis performed in all of them but one female with ME). The median severity score was 10.0 (from 5 to 37) with 22 patients classified as having a mild disease (the score ≤ 8), 77 a moderately severe (9–22) and 4 a severe disease (≥23).

The AM group consisted of 18 patients, 12 male and 6 female, median age 30.5 years, mostly with a mild clinical presentation (M in 16, ME in 2).

There were 19 healthy subjects in the control serum group (12 male, 7 female, median age 46 years) and 12 patients in the control non-inflammatory CSF group (4 male, 8 female, median age 38 years).

### 3.2. Cytokine Concentrations

The median serum concentration was significantly increased for CXCL2, CX3CL1 (both *p* < 0.001), CXCL11 and CCL21 (both *p* < 0.01) in the TBE group compared to healthy controls.

The data on the CSF concentrations in TBE in comparison to control CSF samples and to the simultaneous serum concentrations are summarized in Table 1. Briefly, the CSF median concentrations were significantly higher in TBE than in controls for all chemokines except CXCL2, CCL5 and XCL1, for most of them with a very high statistical significance. The median I_cytokine_ value was on the order of 10 to 1000 for all the upregulated chemokines except CXCL2, CXCL5 and CCL13, supporting their synthesis within CNS. CXCL1, IL-8, CXCL10, CCL2, CCL3, CCL19 and CCL20 were both highly upregulated and had concentration gradients towards CSF in 90–100% of patients, consistent with their chemoattractant function. CCL4, CCL7 and CCL8 had concentration gradients towards CSF in about half of the patients and for several other cytokines, including highly upregulated IL-16 and CXCL13, there was gradient directed towards CSF in a subgroup of patients only (Appendix A).

The serum chemokine concentrations did not differ between TBE and AM groups with the exception of IL-8, which was about 50% higher in AM (*p* < 0.05). The general pattern of the CSF chemokine upregulation was also similar between these groups, but there were higher concentrations of five lymphocyte-attracting chemokines (CXCL11, CCL7, CCL8, CCL13 and CX3CL1) in AM. The difference was the largest for CXCL11, which had its chemotactic gradient reversed towards CSF in the AM group; CCL7 and CCL8 median chemotactic gradients of ~1 in TBE were also evidently positive in AM, meaning that these three chemokines were most likely to play qualitatively different chemotactic role in AM than in TBE (Table A1, Appendix C).

Most of the cytokine concentrations did not differ according to patient age, but some were weakly positively associated with age (most evidently, CXCL9 in CSF) and the IL-16 gradient and CCL19 concentration in CSF were significantly lower in older patients. All the statistically significant associations with age are shown in Table A3, Appendix D.

### 3.3. Correlations with CSF Cellular Parameters

The results of the cytometric examination are summarized in Table A2 in Appendix C. TBE patients presented with a lymphocyte-dominated pleocytosis with significant accompanying populations of neutrophils and monocytes. Within lymphocytes, there was a predominance of CD3+CD4+ Th cells with an addition of CD3+CD8+ Tc/Treg lymphocytes and minor populations of B, NK and double positive CD3+CD4+CD8+ cells. In comparison with blood, there was a fractional enrichment of the CSF population in T CD3+CD4+ cells at the expense of the other cell types and the differences between their percentages in CSF and blood were highly significant, consistent with the strictly regulated leukocyte recruitment into CSF.

There was a trend for higher CSF cell counts in more severe presentations of TBE, reaching statistical significance for T CD3+CD8+, NK and marginally for B cells in MEM compared to other presentations. Contrary to this trend, the lymphocytic pleocytosis and counts of the main lymphocytic populations were significantly lower in TBE than in clinically milder AM.

In the non-parametric analysis, there were multiple correlations of the CSF chemokine concentrations with the CSF cellular parameters, shown as a correlation matrix in Table 2.

The most robust multivariate regression models obtained for total pleocytosis, total lymphocyte count, total T CD3+ as well as T CD3+CD4+, T CD3+CD8+ and B lymphocytes are presented in Table 3, Table 4, Table 5, Table 6, Table 7 and Table 8, respectively.

IL-16 was always recovered as having the strongest association with CSF cytosis and CCL19 was consistently positively associated with lymphocyte counts. A number of other chemokines contributed to pleocytosis, again with a modest role of CXCR3 agonists and CCL4 the only significantly contributing CCR5 agonist. The positive association of CXCL13 with lymphocyte counts was recovered in the T CD3+CD8+ and B cell models. Compared to a non-parametric analysis, there were a number of negative associations, most consistent for IL-8, CXCL11, CCL2 and CX3CL1.

### 3.4. Correlations with the Clinical Presentation

To assess the associations between the chemokine concentrations and the severity, we (1) compared the CSF concentrations between the mild (M) and severe (ME/MEM) presentation of TBE, (2) checked for correlations between the CSF concentrations and the quantitative severity score and, finally, (3) built a multivariate regression model of the clinical severity score as a dependent variable with CSF chemokine concentrations as independent variables. The results of the non-parametric analyses are summed up in Table 9. In brief, the upregulated chemokines could be broadly divided into two subsets: one associating, more or less consistently, with the severe disease and the other not related to severity.

In the multivariate regression analysis, the clinical severity score was best described by a model presented in Table 10, in which CXCL1, CXCL11, CCL17 and CCL20 were recovered as associated with a severe presentation and several chemokines, as associated with a mild presentation, predominantly the ones that were neutral in the non-parametric tests (CCL11, CCL19, CXCL13) and, also, CXCL10, initially appearing as moderately severity-related. IL-16 did not associate with presentation in any analysis.

### 3.5. Correlations with Genetic Background

All the 101 available samples from TBE patients were successfully genotyped for all the studied loci. The genotype distribution in the TBE group is shown in Table A4 in Appendix E. As there was no minor allele detection in rs34334103 (*CXCR3*) and 98% of patients were homozygous for the major G allele in rs2280788 (*CCL5*), these two loci were not further analyzed. Of the remaining loci, the genotypes of rs1799987 in *CCR5*, rs4508917 in *CXCL10* and rs6817952 in *CXCL11* did not associate with any analyzed parameter.

There were some associations between the genotypes and the relevant chemokine concentrations in serum and/or CSF for all four studied chemotactic axes (Figure 1).

There were also some limited associations with the CSF cellular parameters, mostly dependent on small groups of patients homozygous for the minor allele of CCR2 and CX3CR1 genes (Figure 2).

No polymorphism was associated with the clinical severity score.

## 4. Discussion

We have performed a comprehensive study of the chemotactic cytokine expression in *Flavivirus* encephalitis, assessed its relation to the CSF cytometric parameters and the clinical presentation, and attempted a preliminary evaluation of its genetic background. Our results suggest a simultaneous intrathecal upregulation of multiple chemokines not associated directly with the peripheral response, in agreement with previous findings on neuroinflammatory diseases including TBE [33,40,41]. Both the CSF cytometry findings and the chemokine profiles were different in TBE than in a reference group of patients non-TBE meningitis, consistent with the presence of distinct pathogenetic features in *Flavivirus*-related neuroinflammation. Based on our results, we attempted to identify the chemokines most likely to be important in the pathogenesis of human TBE, following several criteria: the likelihood of their intrathecal synthesis, presence of the chemotactic gradient towards CSF, statistical association with CSF cytometric parameters and with the clinical severity.

The chemokines included in our analysis were previously reported in the context of either *Flavivirus* encephalitis or neuroinflammation in general, but the amount of data on their role was variable and we are not aware of any comprehensive analysis assessing them simultaneously in a uniform group of patients. Lepennetier et al. have analyzed a partially overlapping set of 26 chemokines in a diverse population of 75 patients with infectious or autoimmune neuroinflammation, confirming the upregulation and correlation with at least some CSF cellular populations for most of them, but also finding significant variability dependent on etiology [40]. Their group included 10 patients with viral meningitis, but none with *Flavivirus* infection, and the analysis omitted some important mediators included in our study, like CCL4, CXCL10 or IL-8, making the direct comparison difficult [40].

Of the chemokines included in our study design, IL-8 is a ubiquitous chemoattractant upregulated in a wide spectrum of bacterial and viral CNS infections, including TBE [16,33,41]. The related CXCR2 agonists, CXCL1 and CXCL2, were found to be upregulated intrathecally in TBE, with CXCL1 correlating with CSF neutrophil counts and clinical severity [42]. Based on their spectrum of activity, this group of mediators should be indispensable for the neutrophil migration into CNS, but might also contribute to the lymphocyte influx, which we attempted to verify currently [16].

As for the chemokines directed more specifically towards mononuclear cells, two groups have been most extensively studied and hypothesized to play primary roles in the infections with neurotropic *Flavivirus*, which, however, has been questioned by some recent data. CXCR3 agonists (CXCL9, CXCL10, CXCL11) are vividly upregulated intrathecally in murine models of *Flavivirus* encephalitis, exerting pleiotropic chemotactic, pro-inflammatory and immunopathogenic effects [13,23,26,43,44] and CXCL10 is present in a high concentration in CSF of TBE patients [6,31,33]. In human TBE, however, CSF T lymphocyte population is depleted in CXCR3-positive cells, which contradicts the CXCL10/CXCR3 axis driving their migration from periphery [31]. CCR5 agonists (CCL3, CCL4, CCL5) are commonly upregulated in CNS pathology [16,19,43]. The clones established from CSF Th1 lymphocytes from both healthy subjects and patients with autoimmune CNS conditions migrated towards CCL5, CXCL10 and CXCL12, suggesting these chemokines being essential for both housekeeping Th1 lymphocyte circulation within CNS and for their influx in inflammation [2]. Murine models also suggest CCR5 ligands to be crucial drivers of the lymphocyte migration into CNS in *Flavivirus* encephalitis [19,24,25,26] and human genotyping data initially suggested that the susceptibility to TBE was associated with the variability in the gene for CCR5 [29,30]. However, inconsistently with these observations, the CSF T lymphocyte population is only moderately enriched in CCR5-positive cells in human TBE [31] and TBE patients homozygous for the *CCR5Δ32* deletion, lacking a functional CCR5, have a normal clinical presentation and normal CSF cytometric findings [32].

Growing evidence suggests a contribution from additional chemotactic cytokines both in a *Flavivirus* encephalitis and in neuroinfections in general. For example, CXCL12 is upregulated intrathecally in viral but not in bacterial meningitis [40]. The CXCL13 concentration in CSF correlates with the B cell fraction and the intrathecal antibody synthesis in various CNS infections and is characteristically high in Lyme neuroborreliosis [40,41,45]. Increased CSF levels of both these CXC chemokines have been described in TBE [33,45,46]. CCL2 is expressed by human astrocytes infected with TBE virus [44] and creates a concentration gradient towards CSF in TBE [31,33]. In a murine West Nile encephalitis model, a signaling through receptor for CCL2 is indispensable for a monocyte migration into CNS, neuroinfection control and survival [47,48]. CCL7, CCL8, CCL11 and CCL13 have not been studied in TBE so far but are upregulated in human CNS in various conditions including Lyme neuroborreliosis (CCL7, CCL8), bacterial meningitis (CCL11, CCL13), viral meningitis (CCL7, CCL8) and autoimmune encephalitis (CCL11) [40,49]. The CCR7 agonists (CCL19 and CCL21) are involved in autoimmune CNS pathology [2,49]. CCL19 was found indispensable for the development of the specific local immune response in a mouse model of a neurotropic coronavirus disease [50]. Moreover, CCL19 chemotactic gradient towards CSF was detected in patients with neurologic varicella-zoster virus disease [51] and in TBE, where it associated with lymphocytic pleocytosis [33]. The soluble fractalkine was detected by Kastenbauer et al. in CSF in different CNS inflammatory conditions, albeit with no correlation with pleocytosis [52], while in the study by Leppenetier et al., it was associated with the NK cell influx [40]. IL-16 concentration is increased in CSF of patients with viral and bacterial meningitis and correlates with pleocytosis in spite of not creating a detectable chemotactic gradient towards CSF [31,34].

Our current results support a degree of the intrathecal upregulation of most of the cytokines studied but also allow to distinguish a subset most likely involved in TBE pathogenesis. The initial analysis suggested that the main chemotactic factors may be IL-16, which is strongly associated with total pleocytosis and T leukocyte subpopulations, CXCL1 and IL-8 for neutrophils and CXCL13 and CCL19 for lymphocytes. Meanwhile, the role of two groups of chemokines most studied in the context of viral CNS infection, CCR5 agonists and CXCR3 agonists, appeared less evident. The interpretation of these results must be cautious because of the multiple comparisons and intercorrelations between chemokines, but the main observations were both highly statistically significant and biologically relevant. There was a physiologically expectable correlation between IL-16 and CD4-bearing cells, the main CXCR1/CXCR2 agonists and neutrophils, the CCL2 gradient and monocytes and between CCL19 and T lymphocytes. The negative association of CCL2 with lymphocyte counts was consistent with its antagonism with other chemokines in the murine model of JE [53].

The linear regression models confirmed but also partially modified these findings. The strongest and most repeatable correlation with pleocytosis was confirmed for the IL-16 CSF concentration but notably not for the concentration gradient. The association between CCL19 and lymphocytes, the modest role of CXCR3 agonists and the negative association of CCL2 with lymphocyte counts were recovered by the models, while the wide-spectrum chemotactic effect of CXCL13 was not. However, the strong association between CXCL13 with B lymphocyte count and the weaker association with T CD8+ cell count remained, in good agreement with its reported spectrum of activity [16]. Of the CCR5 agonists, CCL4 appeared as a significant chemotactic factor in the total pleocytosis and T CD3+ lymphocyte count models. Besides CCL2, other chemokines were recovered to associate negatively with leukocyte counts, most repeatedly CX3CL1 and neutrophil-associated chemokines in lymphocyte count models. The most robust models included positive and negative contributions from multiple chemokines, with their roles varying for different cell populations, suggesting a complex regulation of the leukocyte migration by a network of molecular stimuli.

Interestingly, IL-16 and, to a smaller extent, selected chemokines, such as CXCL6, correlated with pleocytosis despite only rarely creating a chemotactic gradient towards CSF, suggesting alternative mechanisms of association with leukocyte counts. A similar pattern has been previously observed for IL-16 in TBE and other acute viral CNS infections [33,34]. In viral meningitis, IL-16 reaches top concentrations early and transiently before the 5th day of the disease [34], and its levels could be already decreasing in our patients at the time of CSF collection, resulting in a resolution of the original concentration gradient. However, the correlation of its concentration with the leukocyte populations not expressing CD4 suggests a more complex relationship with pleocytosis than a direct chemotactic effect, which requires further study to elucidate.

The negative correlations of some CSF chemokine concentrations with certain CSF cell counts are counterintuitive but may be biologically feasible. A chemokine could reduce the leukocyte migration acting as a competitive antagonist of a more biologically active chemokine signaling through the same receptor. We have observed pairs of related chemokines with an opposite effect in the lymphocyte count model (CXCL1 and CXCL2, CXCL10 and CXCL11, CCL19 and CCL21). More speculatively, a chemokine could also negatively affect a cell population by promoting a migration of an antagonistic population or, more generally, being involved in the development of other antagonistic arms of the immune response. This has been described for CCL2 in the murine Japanese encephalitis [JE] model by Kim et al., where CCL2 has a net protective immunomodulatory effect and decreases intrathecal expression of other chemokines [53]. Analogously, IL-8, which acts predominantly on neutrophils and was recovered as an antagonist in lymphocyte count models, could have been associated with either the earlier, neutrophil-dominated stage of the CSF infiltration or with a pro-inflammatory, neutrophil-dominated type of the intrathecal response. The negative association of CXCL13 with neutrophils may reflect that it is engaged only in a more mature, specific stage of an immune response. As for CX3CL1, the net effect of its expression may be especially complex, as the membrane form may act as an adhesion molecule for leukocytes and anchor them at the endothelial level, preventing further migration [16,52]. The expression of CX3CL1 on brain endothelia is upregulated in acute neuroinfections [16,52] and could reduce leukocyte migration across blood–brain barriers in our study, reflected by the negative correlation of different lymphocyte subsets with the soluble CX3CL1 derived from the membrane form.

Our results do not identify a single chemotactic factor or a small set of factors uniquely responsible for CSF pleocytosis in TBE. Specifically, they do not confirm the dominant roles of the CCR5 and CXCR3 agonists in lymphocyte migration, in agreement with our previous results [31,32]. Both groups of chemokines were upregulated intrathecally and contributed to the pleocytosis, but their role was modest, with CCL4 having the most significant effect. These findings do not exclude clinically relevant effects of these groups of chemokines. For example, based on the murine model, CXCL10 may determine local leukocyte distribution and thus the type, extent and location of the inflammation within brain parenchyma [23]. On the other hand, CCL19 expressions were associated well with total pleocytosis and lymphocyte count and contributed especially to T CD4+ and T CD8+ count models. As, unlike IL-16, it consistently created a chemotactic gradient towards CSF, it appears as the most important chemoattractant for these lymphocyte populations in TBE, confirming and extending the results of other studies [21,33,51]. As several other chemokines, including CXCL12, CXCL13, CCL7, CCL8 and CCL20, were both highly upregulated intrathecally and recovered in at least some cell count models, they seemed to contribute to the mononuclear cell influx alongside CCL19.

Although the overall expression pattern was similar, we were able to identify quantitative differences in the expression of particular chemokines between mild and severe TBE cases. When analyzed individually, the concentrations of neutrophil-attracting chemokines and selected lymphocyte-attracting ones (CCL2, CCL4, CCL20) were associated very consistently with severity. A group of agonists, including mainly CXCR3,were associated with severity, but in a less evident manner. Other factors, including IL-16 and CCR7agonists CCL19 and CCL21, were evidently not severity-related. The multivariate regression model allowed us to refine these findings. Some of the chemokines that were neutral or weakly associated with severity in non-parametric tests were found to be associated with a milder disease, while some of those associated with a severe presentation remained so in the final model. While there seems to be a trend for a positive association between CNS immunopathology and CSF inflammatory changes, similarly to the animal models [26], our results allow us to delineate the chemokines likely related to the predominantly protective response (CCL11, CCL19, CXCL10 and CXCL13) and to the immunopathogenic one (CXCL1, CCL17 and CCL20). The first group is collectively chemotactic for a mixed mononuclear population, especially including Th1, Tc, Treg and B lymphocytes. The second group includes attractants predominantly for neutrophils and Th17 cells. Interestingly, the chemokine association with neurologic severity, reflecting CNS tissue pathology, was independent from the associations with CSF cell counts. Previously, a correlation between total pleocytosis, the main leukocyte types and the severity of TBE was observed [9,11], but it was always weak and not reproduced in some patient cohorts [54]. Our current results confirm the loose association between quantitative pleocytosis and the severity of CNS tissue pathology by showing that these two aspects of encephalitis may be independently determined at the cytokine level. Highlighting this discrepancy, the patients with AM studied alongside our TBE group had higher median pleocytosis and higher CSF concentration of selected chemokines, although not the ones appearing as the most protective in TBE, despite having a simultaneously milder clinical presentation. The examples of the variable relation between the chemotactic and protective effects of particular cytokines are shown in Figure 3.

Some chemokines, especially CCL19, seem to contribute to pleocytosis while having a protective effect on the CNS, implying the protective role of their target cell populations. The B-cell attractant CXCL13 belongs to that group too, in line with the specific intrathecal humoral response being protective against TBEV both in the animal model [26] and in the clinical data [55,56]. Conversely, the neutrophil-associated chemokine CXCL1 and the Th17-lymphocyte attractant CCL20 were both associated with the influx of leukocytes (respectively, neutrophils and T lymphocytes) and with the clinical severity, suggesting that their target cells may contribute to the immune-mediated pathology of CNS. Interestingly, a protective effect was recovered for the main CXCR3 ligand CXCL10 despite its weak association with pleocytosis, which aligns with animal data showing its critical involvement in the intrathecal response to *Flavivirus* [23,26]. Similarly, a protective effect was also observed for CCL11, which did not create a chemotactic gradient towards CSF. On the other hand, IL-16, despite its strong correlation with pleocytosis, was not associated with clinical presentation. Although it was not evident in our study cohort, there is a tendency for a more severe TBE presentation in older patients [15]. Interestingly, the intrathecal expression of several chemokines tended to be higher in older patients, although the association was weak for most of them, except for CXCL9. On the contrary, the intrathecal expression of IL-16 and of the protective CCL19 was inversely associated with age. This pattern is consistent with a more intensive and pathogenic but less protective in older patients, in agreement with the clinical observations.

Previously, in an independent analysis, we identified CCL20 as a strong prognostic marker of poor neurologic outcomes in TBE [57], which is consistent with the current results regarding this chemokine; it highlights the possibility of using chemokines in the prognostic models of TBE in clinical settings. With a deeper understanding of leukocyte chemotaxis in TBE, the therapeutic strategies could possibly be developed to target the chemotactic signaling pathways, particularly for downregulating the potentially pathogenic CCL20/CCR6 axis.

The associations between the chemokine concentrations and pleocytosis on the one hand and the genetic polymorphisms on the other were dependent on small number of cases but consistent with the phenotypic data and the known activity of the chemokines in question. The increased concentration of the CCL3 associated with *CCR5Δ32*, of CXCL10 with a *CXCR3* variant and of CX3CL1 with a *CX3CR1* variant may all be explained by an increased ligand expression compensatory to a reduced receptor availability or function. The association between the *CCR2* genotype and the CSF monocyte count is consistent with its role in monocyte migration [16,18,47,48]. The minor *CX3CR1* variant was associated both with higher CX3CL1 expression and lower CSF lymphocyte subpopulation counts, in line with a probable antagonistic function of CX3CL1 in controlling lymphocyte migration [16,52]. These results suggest that genetic background may influence chemotactic signaling and, as a consequence, CSF pleocytosis in TBE. They provide no evidence that this variability influences clinical presentation, but such an association may be hypothesized and could be investigated in larger patient groups in the future. The polymorphism for study were selected based on pre-existing knowledge and were all related to signaling axes thought to be important in TBE pathogenesis. However, our phenotypic results suggest that the variability in genes for additional chemokines and receptor may be more clinically relevant. This applies especially to the components of the CCL19/CCR7, CCL20/CCR6 and CXCR13/CXCR5 axes and to chemokines and receptors involved in the neutrophil migration. The role of genetic background in the expression of these factors in TBE warrants further study.

The CSF samples were drawn from the TBE patients in the active neurologic phase of the disease when the CSF pleocytosis was already established. Although the CSF obtained at the onset of neuroinflammation and leukocyte migration into CNS could be more informative, it is rarely available in clinical practice, as patients are rarely hospitalized or undergo lumbar puncture at that early stage. Another limitation is studying CSF leukocytes and not the cells infiltrating CNS tissues, which would more directly probe the inflammatory focus; however, this is rarely possible in a clinical setting. The precision of our statistical analyses may have been lowered by the use of a large number of independent variables in relation to the patient cohort size, a tendency for correlations between the variables, the possibility of non-linearity of some associations and the dependency on individual outliers. We addressed these problems by manually checking and improving the models and excluding problematic variables or cases. Our main goal was not to provide definitive prognostic models of CSF pleocytosis but, rather, to identify the chemotactic cytokines most significantly associated with it, which are worthy of further study. We consider our results reliable in that respect.

As our study focused on TBE as an example of *Flavivirus* encephalitis, the non-TBE group of patients was small, heterogeneous and not intended for in-depth analysis. However, the main results seem tentatively applicable to viral CNS infections in general, as the chemokine profile was similar in both the TBE and non-TBE groups. The main difference in non-TBE patients was a higher intrathecal expression of a few lymphocyte-related cytokines, consistent with higher mononuclear pleocytosis. In light of this, TBE patients, compared to the wider viral meningitis population, appear to have qualitatively similar but relatively weaker and less complex intrathecal chemokine response, which aligns with their relatively low mononuclear pleocytosis.

## 5. Conclusions

We have shown that the CSF pleocytosis in TBE is associated with the activation of multiple, simultaneously co-expressed chemotactic axes. IL-8 and CXCL1 appear to be the main chemokines for neutrophils, while CCL19 may be the main chemotactic factor for lymphocytes. Signaling by CXCL10, CXCL12, CXCL13, CCL4, CCL7, CCL8 and CCL20 may contribute to lymphocyte influx. IL-16 associates strongly with pleocytosis, but its pathogenetic role and causal relationship with pleocytosis remain enigmatic. CX3CL1 may directly inhibit leukocyte migration into CSF. Other chemokines, especially CCL2, may inhibit influx of selected cell populations through direct or indirect antagonism between chemotactic factors. The protective or immunopathogenic effects of the intrathecally expressed chemokines are likely dependent on the detailed spectrum of leukocyte subpopulations they attract into CNS. CCL19, CXCL13, CXCL10 and CCL11 are likely mediators of the protective intrathecal response involving Th1 and B cells, while CXCL1 and CCL20 may contribute to the immune-mediated pathology, including influx of Th17 cells and neutrophilic inflammation. This complexity may explain why total pleocytosis and the counts of the main leukocyte populations are only loosely correlated with the clinical presentation of TBE. Polymorphisms in genes for chemokine ligands and receptors are associated with chemokine expression in TBE and may, hypothetically, influence clinical outcomes.

## Figures and Tables

**Figure 1 cells-14-00490-f001:**
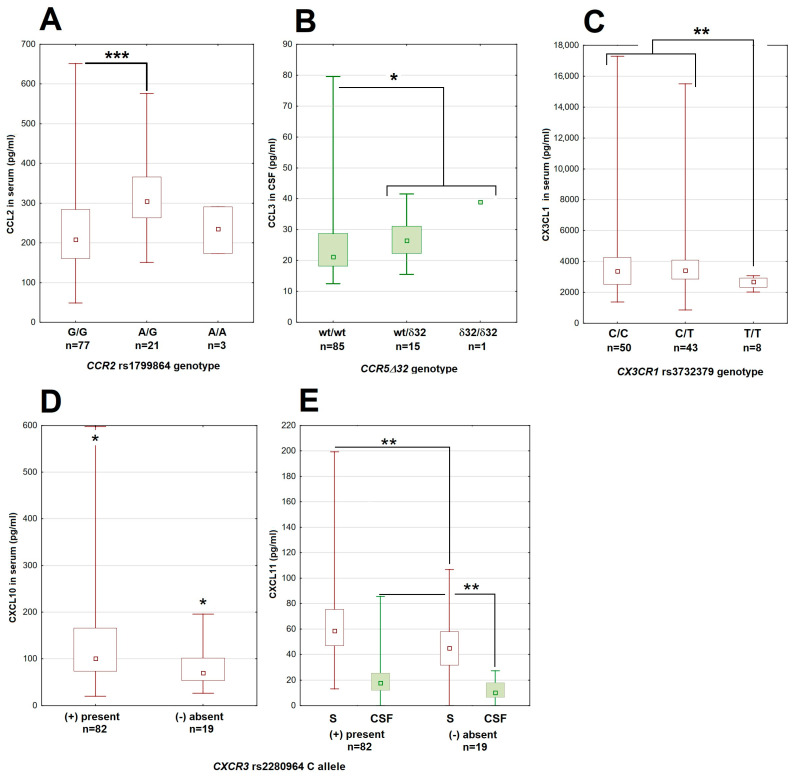
The associations between genetic polymorphisms and chemokine concentrations in patients with tick-borne encephalitis. (**A**) The CCL2 concentration in serum dependent on the genotype in the rs1799864 locus in *CCR2*. (**B**) The higher CCL3 concentration in cerebrospinal fluid in patients with the Δ*32* deletion in *CCR5*. (**C**) The lower concentration of CX_3_CL1 in serum in patients homozygous for the minor T allele in the rs3732379 locus in *CX3CR1*. (**D**,**E**) The lower concentrations of, respectively, CXCL10 in serum and CXCL11 in serum (empty bars) and cerebrospinal fluid (filled bars) in patients lacking the major C allele in the rs2280964 locus in *CXCR3*. The median (square), quartiles (box) and minimum–maximum values (whiskers) are shown. All concentrations expressed in pg/mL; CSF—cerebrospinal fluid, *—significant difference with *p* < 0.05; **—with *p* < 0.01; ***—with *p* < 0.001. The median (square), quartiles (box) and minimum–maximum values (whiskers) are shown.

**Figure 2 cells-14-00490-f002:**
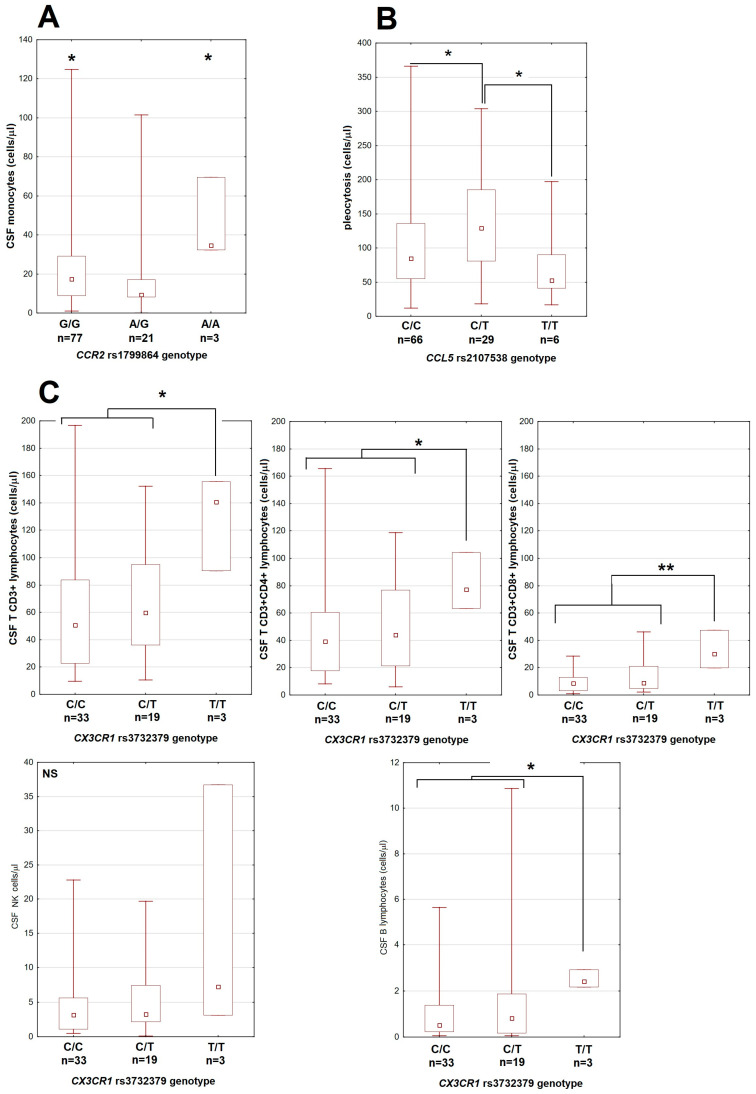
The associations between genetic polymorphisms and cerebrospinal fluid leukocyte counts in patients with tick-borne encephalitis. (**A**) Monocyte count dependent on the genotype in the rs1799864 locus in *CCR2*. (**B**) Total pleocytosis dependent on the genotype in the rs2107538 locus in *CCL5*. (**C**) The lymphocyte subpopulation counts are higher in patients homozygous for T allele in the rs3732379 locus in *CX3CR1*. Upper row from left to right: total T CD3+, T CD3+CD4+ and T CD3+CD8+ lymphocytes; lower left: NK cells, lower right: B lymphocytes (in different scale each). The median (square), quartiles (box) and minimum–maximum values (whiskers) are shown. All cell counts are expressed in cells per μL; CSF—cerebrospinal fluid, *—significant difference with *p* < 0.05; **—with *p* < 0.01; NS—non-significant.

**Figure 3 cells-14-00490-f003:**
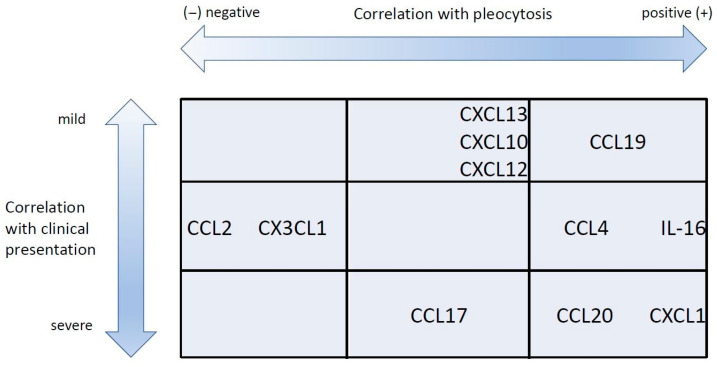
The schematic representation of the association between selected chemokines with cerebrospinal fluid pleocytosis and clinical severity in tick-borne encephalitis. The chemokines associated with a higher pleocytosis but with a milder presentation are grouped at the top right. Those associated with a higher pleocytosis and a more severe presentation are at bottom right. Finally, those associated negatively with pleocytosis are towards the left of the plot. The chemokines listed in the middle row did not associate with the clinical presentation.

**Table 1 cells-14-00490-t001:** The cerebrospinal fluid (CSF) concentrations of chemotactic cytokines in patients with tick-borne encephalitis (n = 103) compared to the paired sera of the same patients and to the non-inflammatory control CSF samples (n = 11).

Cytokine	Concentration in CSF		CSF Concentration Index ^a^ (Median)	CSF/Serum Concentration Gradient ^a^ (Median)	Fraction of Patients with a Concentration Gradient Towards CSF (>1)
	Median (Min–Max) (pg/mL)	Fold Increase Compared to Controls (Statistical Significance)			
IL-16	91.8 (15.4–364.8)	7.8 (*p* < 10^−6^)	43.0	0.46	11%
CXCL1	376.3 (39.8–8185.3)	8.4 (*p* < 10^−6^)	297.9	3.33	90%
CXCL2	18.1 (0–114.9)	1.0 (NS)	2.3	0.02	0%
CXCL5	40.1 (0–448.1)	41.1 ^b^ (*p* < 10^−6^)	3.0	0.03	0%
CXCL6	42.3 (0–862.2)	NC (*p* < 10^−6^)	7.8	0.08	5%
IL-8	347.9 (33.5–5000)	8.5 (*p* < 10^−6^)	6213.2	74.3	100%
CXCL9	384.0 (0–1376.9)	37.2 ^b^ (*p* < 10^−5^)	23.2	0.23	6%
CXCL10	1963.1 (1032.4–3664.4)	10.9 (*p* < 10^−6^)	1927.4	21.7	100%
CXCL11	15.5 (0–85.6)	3.8 (*p* < 0.01)	25.6	0.28	4%
CXCL12	2180.3 (611.3–5275.9)	1.44 (*p* < 0.01)	NA	NA	NA
CXCL13	71.0 (21.3–638.3)	2.8 (*p* < 10^−6^)	43.0	0.46	17%
CCL2	826.9 (274.6–2847.7)	2.0 (*p* < 10^−4^)	330.9	3.71	97%
CCL3	22.1 (12.5–79.6)	1.5 (*p* < 10^−6^)	1556.2	16.66	97%
CCL4	123.7 (70.0–305.3)	1.9 (*p* < 10^−6^)	91.2	0.90	39%
CCL5	0 (0–3881.5)	0.4 ^b^ (*p* < 0.05)	NA	NA	NA
CCL7	55.9 (0–483.5)	34.5 ^b^(*p* < 10^−6^)	101.9	0.99	50%
CCL8	106.1 (10.0–1197.1)	71.3 ^b^ (*p* < 10^−6^)	102.3	1.07	55%
CCL11	26.9 (1.2–60.0)	NC (*p* < 10^−6^)	13.7	0.14	1%
CCL13	3.6 (0–23.0)	2.9 (*p* < 0.05)	2.4	0.03	0%
CCL17	49.1 (0–108.7)	15.7 ^b^ (*p* < 10^−4^)	12.9	0.11	0%
CCL19	1003.1 (120.9–3730.7)	8.1 (*p* < 10^−6^)	1484.7	15.72	100%
CCL20	15.7 (5.8–30.9)	3.1 (*p* < 10^−6^)	245.8	2.84	91%
CCL21	58.2 (0–267.2)	5.8 (*p* < 10^−6^)	14.5	0.16	0%
CX_3_CL1	1434.7 (533.2–4279.1)	1.7 (*p* < 10^−5^)	40.8	0.45	3%
XCL1	14.5 (0–36.3)	0.7 (*p* < 0.05)	9.4	0.10	0%

^a^—calculated as in Section 2. ^b^—mean values compared instead of medians (the median in the control and/or study group equals 0). NC—fold increase non-calculable (cytokine undetectable in controls). NA—non-available (lack of the paired serum measurement for the index and gradient calculation).

**Table 2 cells-14-00490-t002:** The correlations between the cerebrospinal fluid cytokine concentrations and the counts of total leukocytes (pleocytosis) and leukocyte populations in a group of patients with tick-borne encephalitis.

Cytokine	Pleocytosis									
	Total	Neutrophils	Monocytes	Lymphocytes						
				Total	CD3+				NK	B
					Total	CD4+	CD8+	CD4+ CD8+		
IL-16	0.79 (*p* < 10^−10^)	0.24 (*p* < 0.05)	0.40 (*p* < 10^−4^)	0.72 (*p* < 10^−10^)	0.72 (*p* < 10^−9^)	0.73 (*p* < 10^−9^)	0.70 (*p* < 10^−8^)	0.41 (*p* < 0.01)	0.55 (*p* < 10^−4^)	0.61 (*p* < 10^−6^)
CXCL1	0.34 (*p* < 0.001)	0.48 (*p* < 10^−6^)	NS	NS	NS	NS	NS	NS	NS	NS
CXCL5	0.29 (*p* < 0.01)	NS	0.22 (*p* < 0.05)	0.25 (*p* < 0.05)	NS	0.34 (*p* < 0.05)	NS	NS	NS	NS
CXCL6	0.41 (*p* < 10^−4^)	0.33 (*p* < 0.001)	0.31 (*p* < 0.01)	0.28 (*p* < 0.01)	0.31 (*p* < 0.05)	0.38 (*p* < 0.01)	NS	NS	NS	NS
IL-8	0.31 (*p* < 0.01)	0.57 (*p* < 10^−9^)	0.27 (*p* < 0.01)	NS	NS	NS	NS	NS	NS	NS
CXCL9	0.30 (*p* < 0.01)	NS	NS	0.26 (*p* < 0.01)	0.33 (*p* < 0.05)	0.39 (*p* < 0.01)	NS	NS	NS	NS
CXCL10	NS	NS	NS	0.21 (*p* < 0.05)	NS	NS	NS	NS	NS	NS
CXCL11	NS	NS	NS	NS	NS	NS	NS	NS	NS	NS
CXCL12	0.23 (*p* < 0.05)	NS	NS	0.21 (*p* < 0.05)	NS	NS	NS	NS	NS	NS
CXCL13	0.21 (*p* < 0.05)	−0.24 (*p* < 0.05)	NS	0.50 (*p* < 10^−7^)	0.46 (*p* < 0.001)	0.44 (*p* < 0.001)	0.48 (*p* < 0.001)	0.34 (*p* < 0.05)	0.43 (*p* < 0.001)	0.45 (*p* < 0.001)
CCL2	NS	0.45 (*p* < 10^−5^)	NS	−0.29 (*p* < 0.01)	NS	NS	−0.32 (*p* < 0.05)	−0.29 (*p* < 0.05)	−0.34 (*p* < 0.05)	−0.30 (*p* < 0.05)
CCL3	0.22 (*p* < 0.05)	NS	NS	NS	NS	0.28 (*p* < 0.05)	NS	NS	NS	NS
CCL4	0.36 (*p* < 0.001)	0.49 (*p* < 10^−5^)	NS	NS	NS		NS	NS	NS	NS
CCL7	0.32 (*p* < 0.001)	0.21 (*p* < 0.05)	NS	0.22 (*p* < 0.05)	0.29 (*p* < 0.05)	0.37 (*p* < 0.01)	NS	NS	NS	NS
CCL8	0.32 (*p* < 0.001)	0.26 (*p* < 0.01)	NS	0.23 (*p* < 0.05)	NS	0.33 (*p* < 0.05)	NS	NS	NS	NS
CCL11	0.27 (*p* < 0.01)	0.36 (*p* < 0.001)	NS	NS	NS		NS	NS	NS	NS
CCL13	0.36 (*p* < 0.001)	0.34 (*p* < 0.001)	0.29 (*p* < 0.01)	0.19 (*p* < 0.05)	NS	0.28 (*p* < 0.05)	NS	NS	NS	NS
CCL17	NS	NS	NS	NS	NS		NS	NS	NS	NS
CCL19	0.38 (*p* < 0.001)	NS	NS	0.36 (*p* < 0.001)	NS	0.30 (*p* < 0.05)	NS	NS	NS	NS
CCL20	0.41 (*p* < 10^−4^)	0.45 (*p* < 10^−5^)	NS	0.22 (*p* < 0.05)	NS	0.31 (*p* < 0.05)	NS	NS	NS	NS
CCL21	NS	NS	NS	NS	NS	NS	NS	NS	NS	NS
CX_3_CL1	NS	NS	0.26 (*p* < 0.05)	0.21 (*p* < 0.05)	NS	NS	NS	NS	NS	NS

The strength and statistical significance of each correlation are shown in individual cells. NS—non-significant. The concentrations of all the cytokines presented in the table were increased in cerebrospinal fluid of patients with tick-borne encephalitis in comparison with non-inflammatory control CSF.

**Table 3 cells-14-00490-t003:** The multiple linear regression model of the cerebrospinal fluid total pleocytosis in patients with tick-borne encephalitis, dependent on the cerebrospinal fluid concentrations and/or gradients of chemotactic cytokines.

N = 98	R = 0.92485485; R^2^ = 0.85535649; Corrected R^2^ = 0.83493623; F = 41.888; *p* < 0.0000; SE = 33.835				
	β	β * SD	b	b SD	*p*
Free parameter			32.4124	17.8154	0.072
IL-16 concentration	0.992662	0.056708	1.1541	0.0659	<10^−6^
CCL4 concentration	0.209468	0.082368	0.4428	0.1741	<0.05
CXCL6 gradient	0.201127	0.066534	44.1086	14.5913	<0.01
CCL13 gradient	0.175257	0.079667	400.1606	181.9024	<0.05
CCL11 concentration	−0.141349	0.075702	−1.0800	0.5784	0.065
CXCL13 concentration	−0.187228	0.054849	−0.1547	0.0453	<0.001
CCL8 concentration	−0.190527	0.077870	−0.1038	0.0424	<0.05
CX_3_CL1 concentration	−0.313393	0.061724	−0.0411	0.0081	<10^−5^

* The model parameters are presented in the top row, with the parameters for individual independent variables below. The independent variables are ordered from the highest positive to the highest negative standardized regression coefficient value. Contributions from some independent variables did not reach the level of statistical significance, but their removal worsened the overall model parameters. β—standardized regression coefficient. b—regression coefficient. SD—standard deviation. SE—standard estimation error.

**Table 4 cells-14-00490-t004:** The multiple linear regression model of the cerebrospinal fluid total lymphocyte counts in patients with tick-borne encephalitis, dependent on the cerebrospinal fluid concentrations and/or gradients of chemotactic cytokines.

N = 90	R = 0.90991647; R^2^ = 0.82794798; Corrected R^2^ = 0.79402223; F = 24.405; *p* < 0.0000; SE = 25.145				
	β	β * SD	b	b SD	*p*
Free parameter			6.129	17.52460	NS
IL-16 concentration	0.931665	0.071108	0.697	0.05317	<10^−6^
CCL20 concentration	0.157384	0.075028	1.656	0.78944	<0.05
CCL19 concentration	0.145336	0.076495	0.014	0.00757	0.062
CXCL1 gradient	0.130906	0.060285	1.827	0.84149	<0.05
CCL3 gradient	0.129082	0.059048	0.273	0.12468	<0.05
CXCL10 concentration	0.104640	0.060787	0.010	0.00572	0.089
CXCL12 concentration	0.094197	0.063804	0.006	0.00383	0.144
CX_3_CL1 concentration	−0.102323	0.079759	−0.010	0.00793	0.204
CCL13 concentration	−0.131715	0.069826	−2.264	1.20026	<0.063
CCL2 concentration	−0.159985	0.059360	−0.019	0.00692	<0.01
CXCL2 gradient	−0.164208	0.054894	−258.784	86.50934	<0.01
CXCL11 concentration	−0.170261	0.075926	−0.704	0.31380	<0.05
IL-8 gradient	−0.249295	0.067334	−0.193	0.05218	<0.001
CCL21 concentration	−0.278474	0.066894	−0.413	0.09916	<10^−4^

* The model parameters are presented in the top row, with the parameters for individual independent variables below. The independent variables are ordered from the highest positive to the highest negative standardized regression coefficient value. Contributions from some independent variables did not reach the level of statistical significance, but their removal worsened the overall model parameters. β—standardized regression coefficient. b—regression coefficient. SD—standard deviation. SE—standard estimation error.

**Table 5 cells-14-00490-t005:** The multiple linear regression model of the cerebrospinal fluid T CD3+ lymphocyte counts in patients with tick-borne encephalitis, dependent on the cerebrospinal fluid concentrations and/or gradients of chemotactic cytokines.

N = 48	R = 0.94170363; R^2^ = 0.88680572; Corrected R^2^ = 0.84799626; F = 22.850; *p* < 0.00000; SE = 17.615				
	β	β * SD	b	b SD	*p*
Free parameter			11.912	12.97947	NS
IL-16 concentration	0.635714	0.101137	0.442	0.07028	<10^−6^
CCL4 concentration	0.421289	0.124461	0.516	0.15250	<0.01
CXCL6 concentration	0.311816	0.106263	0.088	0.02990	<0.01
CXCL9 gradient	0.258454	0.083191	6.419	2.06620	<0.01
CCL19 concentration	0.237197	0.088684	0.018	0.00683	<0.05
CXCL12 concentration	0.172794	0.076303	0.009	0.00392	<0.05
CXCL2 concentration	−0.217161	0.080685	−0.377	0.14004	<0.05
CX_3_CL1 concentration	−0.228741	0.112061	−0.015	0.00756	<0.05
CCL21 gradient	−0.249035	0.075813	−137.665	41.90870	<0.01
CCL2 concentration	−0.283288	0.084089	−0.022	0.00646	<0.01
IL-8 gradient	−0.296352	0.079352	−0.125	0.03338	<0.001
CXCL11 concentration	−0.406704	0.104557	−1.047	0.26924	<0.001

* The model parameters are presented in the top row, with the parameters for individual independent variables below. The independent variables are ordered from the highest positive to the highest negative standardized regression coefficient value. One outlying case was excluded from the model. β—standardized regression coefficient. b—regression coefficient. SD—standard deviation. SE—standard estimation error.

**Table 6 cells-14-00490-t006:** The multiple linear regression model of the cerebrospinal fluid T CD3+CD4+ lymphocyte counts in patients with tick-borne encephalitis, dependent on the cerebrospinal fluid concentrations and/or gradients of chemotactic cytokines.

N = 50	R = 0.93254379; R^2^ = 0.86963793; Corrected R^2^ = 0.83621175; F = 26.017 *p* < 0.00000; SE = 13.326				
	β	β * SD	b	b SD	*p*
Free parameter			4.50770	8.053756	NS
IL-16 concentration	0.667840	0.087971	0.34303	0.045186	<10^−6^
CCL19 gradient	0.356698	0.085453	2.06708	0.495203	<0.001
CCL11 gradient	0.347663	0.079106	37.30562	8.488352	<10^−4^
CXCL9 concentration	0.219668	0.111690	0.01887	0.009594	0.056
CXCL6 concentration	0.177171	0.087272	0.03695	0.018199	<0.05
CCL20 concentration	0.155866	0.078334	0.29729	0.149407	0.054
CX_3_CL1 concentration	−0.153387	0.100032	−0.00751	0.004898	0.133
CCL2 gradient	−0.221737	0.079749	−1.83547	0.660140	<0.01
IL-8 gradient	−0.377169	0.084266	−0.11766	0.026288	<10^−4^
CXCL11 concentration	−0.504231	0.138359	−0.96561	0.264960	<0.001

* The model parameters are presented in the top row, with the parameters for individual independent variables below. The independent variables are ordered from the highest positive to the highest negative standardized regression coefficient value. Contribution from one independent variable did not reach the level of statistical significance, but its removal worsened the overall model parameters. One outlying case was excluded from the model. β—standardized regression coefficient. b—regression coefficient. SD—standard deviation. SE—standard estimation error.

**Table 7 cells-14-00490-t007:** The multiple linear regression model of the cerebrospinal fluid T CD3+CD8+ lymphocyte counts in patients with tick-borne encephalitis, dependent on the cerebrospinal fluid concentrations and/or gradients of chemotactic cytokines.

N = 48	R = 0.87469696; R^2^ = 0.76509478; Corrected R^2^ = 0.70945933; F = 13.752; *p* < 0.00000; SE = 5.3875				
	β	β * SD	b	b SD	*p*
Free parameter			1.47765	3.811226	NS
IL-16 concentration	0.669276	0.121762	0.10289	0.018720	<10^−5^
CCL19 concentration	0.281524	0.096631	0.00483	0.001657	<0.01
CXCL12 concentration	0.267368	0.114724	0.00306	0.001313	<0.05
CXCL13 concentration	0.233792	0.139978	0.02214	0.013255	0.104
CCL4 concentration	0.230491	0.168368	0.06151	0.044931	0.179
CXCL5 concentration	0.196506	0.163790	0.02784	0.023206	0.238
CX_3_CL1 gradient	−0.158132	0.099245	−4.95896	3.112283	0.119
CCL2 concentration	−0.400523	0.108066	−0.00678	0.001830	<0.001
CCL3 concentration	−0.759196	0.219224	−0.65143	0.188106	<0.01

* The model parameters are presented in the top row, with the parameters for individual independent variables below. The independent variables are ordered from the highest positive to the highest negative standardized regression coefficient value. Contributions from some independent variables did not reach the level of statistical significance, but their removal worsened the overall model parameters. One outlying case was excluded from the model. β—standardized regression coefficient. b—regression coefficient. SD—standard deviation. SE—standard estimation error.

**Table 8 cells-14-00490-t008:** The multiple linear regression model of the cerebrospinal fluid B CD19+ lymphocyte counts in patients with tick-borne encephalitis, dependent on the cerebrospinal fluid concentrations and/or gradients of chemotactic cytokines.

N = 47	R = 0.88595594; R^2^ = 0.78491793; Corrected R^2^ = 0.74631345; F = 20.332; *p* < 0.00000; SE = 0.57915				
	β	β * SD	b	b SD	*p*
Free parameter			1.455959	0.337757	<0.001
IL16 concentration	0.744771	0.108984	0.013271	0.001942	<10^−6^
CXCL13 concentration	0.452551	0.115015	0.004901	0.001246	<0.001
CXCL6 gradient	0.449317	0.114498	1.189121	0.303020	<0.001
CCL19 gradient	0.296957	0.091561	0.019546	0.006027	<0.01
CXCL12 concentration	−0.266704	0.088663	−0.000358	0.000119	<0.01
IL8 gradient	−0.405722	0.104101	−0.004308	0.001105	<0.001
CX_3_CL1 concentration	−0.701149	0.117725	−0.001198	0.000201	<10^−5^

* The model parameters are presented in the top row, with the parameters for individual independent variables below. The variables are ordered from the strongest positive to the strongest negative correlation. Two outlying cases were excluded from the model. β—standardized regression coefficient. b—regression coefficient. SD—standard deviation. SE—standard estimation error.

**Table 9 cells-14-00490-t009:** The cerebrospinal fluid (CSF) median concentrations and concentration gradients of chemotactic cytokines in tick-borne encephalitis patients dependent on the clinical presentation.

Cytokine	Concentration (pg/mL)		CSF/Serum Concentration Gradient		Correlation of the Concentration with the Clinical Severity Score ^a^
	M (n = 53)	ME/MEM (n = 50)	M (n = 53)	ME/MEM (n = 50)	
IL-16	83.1	98.33	0.46	0.43	NS
CXCL1	300.9 **	444.56 **	2.96**	3.92 **	0.33 (*p* < 0.001)
CXCL2	18.6	18.11	0.02	0.02	NS
CXCL5	35.0 *	48.85 *	0.03 *	0.04 *	NS
CXCL6	36.4 **	50.00 **	0.07 **	0.11 **	0.26 (*p* < 0.01)
IL-8	282.2 ***	459.69 ***	59.39 **	101.00 **	0.40 (*p* < 10^−4^)
CXCL9	361.8 *	460.58 *	0.20 *	0.27 *	0.22 (*p* < 0.05)
CXCL10	1836.4 **	2426.14 **	23.19	18.59	0.22 (*p* < 0.05)
CXCL11	13.2 *	18.88 *	0.24 ***	0.41 ***	NS
CXCL12	2032.3	2338.77	not calculated ^b^		NS
CXCL13	75.1	64.27	0.46	0.47	NS
CCL2	645.0 **	1023.20 **	2.99 **	4.4 **	0.25 (*p* < 0.05)
CCL3	21.2	24.18	34.28	8.02	NS
CCL4	114.9 **	135.92 **	0.85 *	0.99 *	0.23 (*p* < 0.05)
CCL5	0.0	0.00	not calculated ^b^		NS
CCL7	51.5	69.75	0.91	1.16	NS
CCL8	99.8	113.97	0.97	1.19	NS
CCL11	23.7 **	29.74 **	0.12	0.16	NS
CCL13	2.8	3.66	0.02 *	0.03 *	NS
CCL17	46.9	54.79	0.11	0.12	NS
CCL19	995.0	1014.21	13.57	17.36	NS
CCL20	14.8 **	19.32 **	2.63 *	3.16 *	0.31 (*p* < 0.01)
CCL21	54.6	60.82	0.16	0.16	NS
CX_3_CL1	1427.2	1434.74	0.43	0.47	NS
XCL1	14.5	14.53	0.10	0.10	NS

M—meningitis; ME—meningoencephalitis; MEM—meningoencephalomyelitis. ^a^—severity score calculated according to Bogovič et al. [36]. ^b^—no serum measurements available for comparison with CSF. *—significant difference between the M and combined ME/MEM groups with *p* < 0.05; **—with *p* < 0.01; ***—with *p* < 0.001.

**Table 10 cells-14-00490-t010:** The multiple linear regression model of the Bogovič severity score of tick-borne encephalitis [36] dependent on the cerebrospinal fluid concentrations and/or gradients of chemotactic cytokines.

N = 90	R = 0.84146836; R^2^ = 0.70806900; Corrected R^2^ = 0.67522676; F = 21.560; *p* < 0.00000; SE = 3.0387				
	β	β * SD	b	b SD	*p*
Free parameter			12.69167	1.379226	<10^−6^
CXCL1 concentration	0.653704	0.077975	0.00309	0.000368	<10^−6^
CCL20 concentration	0.464941	0.093631	0.44239	0.089090	<10^−5^
CXCL11 gradient	0.426848	0.067876	4.34984	0.691695	<10^−6^
CCL17 gradient	0.159246	0.074923	6.01605	2.830468	<0.05
CXCL12 concentration	−0.073457	0.069967	−0.00042	0.000399	0.296929
CXCL13 concentration	−0.181374	0.066997	−0.00931	0.003438	<0.01
CXCL10 gradient	−0.241174	0.069927	−0.08663	0.025117	<0.001
CCL19 gradient	−0.253881	0.071612	−0.09557	0.026958	<0.001
CCL11 concentration	−0.498870	0.101186	−0.24135	0.048953	<10^−5^

* The model parameters are presented in the top row, with the parameters for individual independent variables below. The independent variables are ordered from the highest positive to the highest negative standardized regression coefficient value. Contribution from one independent variable did not reach the level of statistical significance, but its removal worsened the overall model parameters. β—standardized regression coefficient. b—regression coefficient. SD—standard deviation. SE—standard estimation error.

## Data Availability

The datasets used and/or analyzed during the current study are available from the corresponding author upon reasonable request.

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
