# Peer review of "The Role of Chemotactic Cytokines in Tick-Borne Encephalitis"

_cells, 2025, doi:10.3390/cells14070490_

Round 1

Reviewer 1 Report

Comments and Suggestions for Authors

The manuscript I've received for review, entitled "The Role of Chemotactic Cytokines in Tick-Borne Encephalitis", presents a comprehensive study of the involvement of chemotactic cytokines in tick-borne encephalitis (TBE). The research is well structured, with clear hypotheses and robust methodology, particularly with regard to multiplex analysis of cytokine expression and flow cytometry for immune cell characterisation. Overall, the paper is very good and the results are very promising. The manuscript makes a valuable contribution to the understanding of the immunopathology of TBE. However, some minor flaws relating to formatting, clarity and data interpretation need to be addressed before the manuscript can be considered for publication.

The abstract should avoid excessive technical jargon and be made accessible to a broader audience.

L 26, 27 – italics in abbreviation of chemokines

L 42 – space in and

L 86 – reference with year

L 208 – space between P and < (you do not have space in the other ones, so use the same type of format in all of them)

L 257 - what does the last sentence refer to?

Tables and figures should be formatted according to MDPI guidelines; currently, some tables lack proper headers and alignment, the headings are inconsistent in font size and bolding (e.g. Table 3 and 4 legend).

While the results provide important insights into chemokine involvement in TBE, discussion is very extensive and somewhat overwhelming, also regarding general issues of chemotactic cytokine expression mechanisms. Perhaps it would be worth moving some of this general information to the introduction section, so as not to be overloaded with too much detail and to focus on discussing the results of these studies with the results of other researchers.

Author Response

The abstract should avoid excessive technical jargon and be made accessible to a broader audience.

The abstract has been rewritten to make it more clear.

L 26, 27 – italics in abbreviation of chemokines

These are actually names of genes for chemokines, so the italics use is correct. It should be apparent in the revised version of the abstract.

L 42 – space in and

This has been corrected

L 86 – reference with year

Has been corrected

L 208 – space between P and < (you do not have space in the other ones, so use the same type of format in all of them)

Has been corrected

L 257 - what does the last sentence refer to?

The sentence has been removed

Tables and figures should be formatted according to MDPI guidelines; currently, some tables lack proper headers and alignment, the headings are inconsistent in font size and bolding (e.g. Table 3 and 4 legend).

Has been checked and corrected (tables 3-5)

While the results provide important insights into chemokine involvement in TBE, discussion is very extensive and somewhat overwhelming, also regarding general issues of chemotactic cytokine expression mechanisms. Perhaps it would be worth moving some of this general information to the introduction section, so as not to be overloaded with too much detail and to focus on discussing the results of these studies with the results of other researchers.

The general information on chemokines from the 2nd to 4th paragraph of the discussion was moved to introduction into its 2nd paragraph, which is now reedited and expanded (ll. 60-87). The text left in the discussion deals mostly with chemokines in neuroinflammation – it was also re-edited. It consists of 4 paragraphs now (2nd-5th of the Discussion, ll. 518-580), the beginning of the following paragraph was also slightly re-edited.

We hope this layout will be more clear and informative.

We added one new reference to a paper on CXCL1 and CXCL2 in TBE and the order of several references was changed.

Reviewer 2 Report

Comments and Suggestions for Authors

To present the relevance of chemokines in the pathogenesis of TBE.
It is necessary to specify whether negative controls were implemented to detect chemokines to prevent experimental bias.
It is necessary to discuss whether individual factors such as age or comorbidity can affect the variability in the expression of chemokines.
To examine the possibility that chemokines have a predictive value in the severity or progression of the disease.
To incorporate a section on the possible use of chemokines as biomarkers or therapeutic targets.
To strengthen the clinical context, address the similarities and differences with other viral infections of the central nervous system.

Author Response

To present the relevance of chemokines in the pathogenesis of TBE

The chemokine relevance stems from their role in driving leukocyte migration into CNS and CSF in inflammation and our research is focused on elucidating this process. Currently, the large parts of the introduction (esp. ll. 67-80) and discussion (especially ll. 518-580) have been rewritten and we hope this is now presented in a more clear way.

It is necessary to specify whether negative controls were implemented to detect chemokines to prevent experimental bias.

Yes, the negative controls for both serum and cerebrospinal fluid were evaluated as mentioned in the second paragraph of 2.1. section. The fold increase of a median chemokine concentration in CSF compared to controls is listed in Table 1.

 It is necessary to discuss whether individual factors such as age or comorbidity can affect the variability in the expression of chemokines.

The patients selected for the study had no other active central nervous system infection or immune-mediated disease, so the significant comorbidity influence on the chemokine synthesis within CNS was not considered – we add explanation on this in the first paragraph of 2.1 section (ll. 118-120).

We have added information on the associations with age as a separate paragraph at the end of 3.2 section (ll. 322-325) and additional table A3 in a new Appendix C (please note that the Appendix order changed as a result and a previous Appendix C table A3 is now Appendix D table A4). We also add a passage discussing it at the end of the paragraph below Figure 3 in Discussion (ll. 707-713). One new reference was added in this last passage.

To examine the possibility that chemokines have a predictive value in the severity or progression of the disease.

To incorporate a section on the possible use of chemokines as biomarkers or therapeutic targets.

There has been little data published regarding the two points mentioned above. We add a paragraph addressing these questions in Discussion, with an added reference to our previous publication on CCL20 as a prognostic marker in TBE (ll. 714-720, ref. 57). To put this in context, we ad a sentence about current therapeutic difficulties in TBE at the end of the first paragraph of introduction (ll. 52-54).

To strengthen the clinical context, address the similarities and differences with other viral infections of the central nervous system.

We compare chemokine expression in TBE to a reference group of patients with mild viral meningitis in table A1. We add a discussion of these results in ll. 681-685.

The paragraphs 2nd to 5th of the Discussion (ll. 518-580) are now thoroughly rewritten and include (especially paragraph 2) more information and hopefully more clearly presented on the previous findings regarding chemokine expression in TBE and other neuroinfections.

Round 2

Reviewer 2 Report

Comments and Suggestions for Authors

The article is ready for publication. The authors have addressed all my comments.